# Laser-Induced Breakdown Spectroscopy for Determination of Spectral Fundamental Parameters

**Sabrina Messaoud Aberkane** [1], **Ali Safi** [2], **Asia Botto** [3], **Beatrice Campanella** [3],
**Stefano Legnaioli** [3], **Francesco Poggialini** [3,4], **Simona Raneri** [3], **Fatemeh Rezaei** [5]
**and Vincenzo Palleschi** [3,*]

1   Division des Milieux Ionisés et Lasers, Centre de Développement des Technologies Avancées,
    Baba Hassen 16081, Algeria; aberkanesabrina16@gmail.com
2   Laser and Plasma Research Institute, Shahid Beheshti University, G. C., Evin, Tehran 11369, Iran;
    a_safi@sbu.ac.ir
3   Applied and Laser Spectroscopy Laboratory, Institute of Chemistry of Organometallic Compounds of CNR,
    56124 Pisa, Italy; asia.botto@gmail.com (A.B.); beatrice.campanella@pi.iccom.cnr.it (B.C.);
    stefano.legnaioli@cnr.it (S.L.); francesco.poggialini@pi.iccom.cnr.it (F.P.); simona.raneri@pi.iccom.cnr.it (S.R.)
4   Classe di Scienze, Scuola Normale Superiore, 56126 Pisa, Italy
5   Department of Physics, K. N. Toosi University of Technology, Tehran 11369, Iran; fatemehrezaei@kntu.ac.ir
*   Correspondence: vincenzo.palleschi@cnr.it

**Abstract:** In this review, we report and critically discuss the application of LIBS for the determination of plasma-emission fundamental parameters, such as transition probabilities, oscillator strengths, Stark broadening and shifts, of the emission lines in the spectrum. The knowledge of these parameters is of paramount importance for plasma diagnostics or for quantitative analysis using calibration-free LIBS methods. In the first part, the theoretical basis of the analysis is laid down; in the second part, the main experimental and analytical approaches for the determination by LIBS of the spectral line spectroscopic parameters are presented. In the conclusion, the future perspectives of this kind of analysis are discussed.

**Keywords:** LIBS; plasma; spectroscopy; transition probabilities; oscillator strengths; Stark broadening; Stark shifts

## 1. Introduction

Laser-induced breakdown spectroscopy (LIBS) [1] is a spectroanalytical technique that is attracting a wide interest for its applications to in situ industrial processes [2–5], environmental analysis [6–8], the study of geological materials [9–11] and cultural heritage objects [12–14] and planetary exploration [15–17], to mention a few. At the laboratory level, excellent results have been obtained in microanalysis and elemental mapping [18–20]. The advantages with respect to other more assessed microanalytical techniques were mainly exploited especially in geological [21–23], cultural heritage [24–26] and biomedical applications [27–29].

Another important field where LIBS analysis has been applied, although sometimes neglected with respect to the large variety of practical applications, is the study of the complex processes occurring in the plasma during its formation and evolution [30]. This kind of fundamental approach has involved the study of the process of laser ablation [31,32], the generation of shock waves [33,34] and the analysis of the emitted radiation [35].

This latter phenomenon is the base of the analytical application of LIBS. Consequently, studies about the fundamental parameters of the plasma radiation (transition probabilities or, equivalently, oscillator strengths [36] and Stark broadening and shift coefficients [37,38]) are of paramount importance

for the development of LIBS as an analytical technique, as well as for the more general studies on the process of plasma emission.

These parameters are traditionally determined in plasma physics studies involving pulsed or continuous plasma sources (linear pulsed discharges, arcs) and conventional emission, absorption or fluorescence spectroscopic techniques. All these methods are considerably more complex with respect to LIBS, which has the further advantage, with respect to traditional methods, of working on potentially all the kinds of materials, conducting or insulating, in gas, liquid or solid form. However, the simplicity of the experimental setup is compensated, in LIBS, by the complex physicochemical processes occurring in the laser-induced plasma. LIBS plasmas are characterized by fast spatial and temporal changes; they are by nature transient, inhomogeneous and out of thermal equilibrium. Therefore, a deep knowledge of the phenomena occurring inside the laser–plasma and their effect on the emitted radiation is needed for a correct interpretation of the experimental results.

In this review, we focus our attention on the main models of plasma radiation and on the different experimental strategies that may be used for the determination of the fundamental spectral parameters using LIBS.

## 2. Plasma Models

The LIBS plasmas are difficult to study because of the multitude of complex phenomena and interactions that characterize them. Sticking momentarily to LIBS plasma generated on solid samples, one should consider that one of the peculiar characteristics of LIBS is the key role of the laser in both sampling (through laser ablation, in solids) and excitations of the ablated material.

While this characteristic is at the basis of all the practical advantages of LIBS (simplicity, possibility of remote analysis, no sample treatment, etc.) the intertwining of laser–sample, laser–plasma, plasma–environment, laser–environment, interactions is extremely complex and difficult to model. Changing the main parameters of the laser beam (pulse length and shape, energy, focusing conditions) and of the environment (composition and density of the environmental gas) may result in large changes in the radiation emitted by the plasma. Moreover, the LIBS plasmas evolve in time, expanding in the surrounding environment. Consequently, spatial gradients of the plasma parameters (atomic and electron number density, electron temperature) may occur, which must be considered in the analysis of the LIBS spectra. Different emission regimes may take place at different times after the onset of the plasma, and the emitted radiation would interact with the plasma itself.

An approximation which is often used for modeling the plasma and its emission is the local thermal equilibrium (LTE) approximation [39].

In the framework of LTE approximation, it is assumed that all the processes in the plasma (ionization–recombination, excitation–deexcitation, etc.) would be at the equilibrium at the same temperature T (eV), except for the radiation, in which the absorption process would not balance the emission.

The equilibrium between ionization and recombination is described by the Saha–Eggert equation:

$$\frac{n^{II}}{n^{I}} = \frac{2}{n_e} \Lambda^{-3} \frac{U^{II}(T)}{U^{I}(T)} e^{-E_{ion}/k_B T} \tag{1}$$

$E_{ion}$ is the (first) ionization energy of the element, $n^{I}$ represents the number density of the neutral species, $n^{II}$ is the number density of the ionic species, $n_e$ is the electron number density and $k_B$ is the Boltzmann constant. $\Lambda$ (cm) is the average thermal De Broglie wavelength of the electrons in the plasma, defined as:

$$\Lambda = \sqrt{\frac{2\pi\hbar^2}{m_e k_B T}} \tag{2}$$

where $m_e$ is the electron mass and $\hbar$ is the reduced Planck constant and $U^{I}(T)$ and $U^{II}(T)$ are the partition functions of the neutral/ionic species.



In LIBS, ionization stages higher than the first are seldom observed, therefore one can safely assume that $n^I + n^{II} = n_{el}$. While the concentration $n_{el}$ of the element can be considered as constant (excluding the possible formation of molecular species in the plasma), the concentration of the species varies during the evolution of the plasma, following the variation of the plasma temperature and electron number density.

The excitation–deexcitation equilibrium is described by Equation (3), which links the number of photons emitted per second in a given transition between two levels of energy $E_k$ and $E_i$, respectively (central wavelength $\lambda_0 = hc/(E_k - E_i)$) to the number density of the corresponding species $n^a$, at the same temperature $T$:

$$I_0 = Fn^a g_k A_{ki} \frac{e^{-\frac{E_k}{k_B T}}}{U^a(T)} \qquad a = \begin{cases} I & \text{for neutral lines} \\ II & \text{for ionized lines} \end{cases} \tag{3}$$

$g_k$ is the degeneracy of the upper level and $A_{ki}$ (s$^{-1}$) is the transition probability.

$I_0$ (photons per second) is thus proportional, through a factor F depending on the acquisition geometry, to the population of the upper level $E_k$ (eV) of the transition, given by the Boltzmann equation:

$$n^a(E_k) = n^a g_k \frac{e^{-\frac{E_k}{k_B T}}}{U^a(T)} \tag{4}$$

times the transition probability $A_{ki}$. As discussed before, in LIBS the process of emission–absorption of the radiation is not at thermal equilibrium (and it should not be, otherwise we would have only the continuous black-body radiation). However, a certain degree of absorption of the plasma radiation by the plasma itself (self-absorption) is physically unavoidable. In fact, according to the Einstein theory, the transition probability $A_{ki}$ is linked to the absorption coefficient B by the relation:

$$A_{ki} = B \frac{8\pi h}{\lambda_0^3} \tag{5}$$

The critical parameter for the validity of Equations (1) and (3) is the ratio between the characteristic times of the radiative processes ($A_{ki}^{-1}$) and the time needed for reaching the equilibrium in the ionization–recombination and excitation–deexcitation processes. To be in LTE conditions, the former should be much longer than the latter.

The ionization–recombination and excitation–deexcitation equilibrium occur because of the collisional processes between the electrons (light and fast) and the atoms/ions (heavy and slow). The electron number density is thus a key factor for a plasma to be or not in LTE conditions.

When the plasma can be considered as homogeneous and stationary, for being in LTE condition is sufficient to have:

$$n_e > 1.6 \cdot 10^{12} \, T^{\frac{1}{2}} \, (\Delta E_{max})^3 \tag{6}$$

where $\Delta E_{max}$ is the largest energy gap between consecutive levels. The numerical coefficient is calculated assuming the electron number density is expressed in cm$^{-3}$, the temperature in K and $\Delta E_{max}$ in eV.

Equation (6) is known as the McWhirter condition [40], and is in general calculated considering for $\Delta E_{max}$ the gap between the ground state and the first excited level, since the higher levels are more closely spaced.

Although in the general case, the McWhirter condition is only necessary, but not sufficient, in the case of homogeneous and stationary plasma, it is also sufficient.

One should note again that the LTE conditions could be incompatible with the validity of Equation (3), because of the self-absorption in the plasma.

In this case, when self-absorption is not negligible (optically thick plasma), Equation (3) should be rewritten considering the escape of the radiation from an absorbing (and emitting) plasma [41].

In LIBS, the emission line profile is usually well described by a Lorentzian function $L(\lambda)$, with the maximum at $\lambda_0$ and the full-width at half-maximum (FWHM) equal to $\Delta\lambda_0$, represented by the formula:

$$L(\lambda) = I_0/\pi \frac{\Delta\lambda_0/2}{(\lambda - \lambda_0)^2 + \left(\frac{\Delta\lambda_0}{2}\right)^2} \tag{7}$$

In Equation (7), $I_0$ is defined as in Equation (3), representing the integral of the Lorentzian function. Equation (7) describes the emission of a plasma in which the photons are not substantially reabsorbed by the emitting atoms, a situation that is likely to occur when the number concentration of the emitters in the plasma is very low. In most of the cases, however, the propagation of the radiation through the plasma would produce a deformation of the line shape with respect to the original Lorentzian profile, characterized by a reduction of the peak intensity and a broadening of the observed line shape. The observed line shape of the emission line, after traveling a length $L$ in the plasma, can be expressed by the relation:

$$I(\lambda) = S(\lambda)\left(1 - e^{-k(\lambda)l}\right) \tag{8}$$

where, neglecting the stimulated emission,

$$k(\lambda) = \frac{\lambda_0^4}{8\pi c} A_{ki} g_k \frac{n_i^a}{g_i} L(\lambda) \tag{9}$$

$n_i{}^a$ is the population of the lower level of the transition, given by the Boltzmann equation as:

$$n_i^a = n^a \frac{e^{-\frac{E_i}{k_B T}}}{U^a(T)} \tag{10}$$

In Equation (8) $S(\lambda_0)$ represents the Planck black-body radiation function. The product $\tau(\lambda) = k(\lambda)l$ is called the optical depth (or optical thickness) of the transition [42]. When $\tau(\lambda)$ is << 1, it's easy to check that Equation (8) coincides with Equation (3), but, since the effect of self-absorption is higher near the maximum of the emission, with its increase the line shape becomes progressively distorted, as shown in Figure 1.

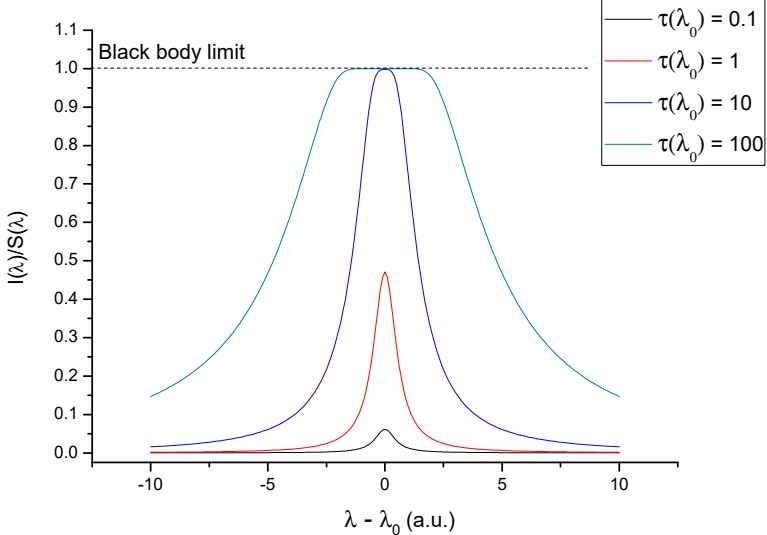

**Figure 1.** Effect of self-absorption on the line shape (numeric calculation).

For resonant lines ($E_i = 0$) the lower level of the transition is generally very populated and, depending on the value of $n^a$, a strong self-absorption can be observed.

Strong self-absorption would also mean a nonlinear relation between observed intensity and the number density of the corresponding species (see Figure 1). In the limit of very strong self-absorption, the line profile flattens out at the Black-body limit, and the intensity becomes independent on the species concentration.

If the plasma is not homogeneous (one can realistically consider a situation where the temperature in the plasma periphery is lower than in the center, for example) the emitted radiation can suffer from a further self-absorption, which affects only the central part of the line. This phenomenon is called self-reversal because it produces a dip in the peak of the line (see Figure 2).

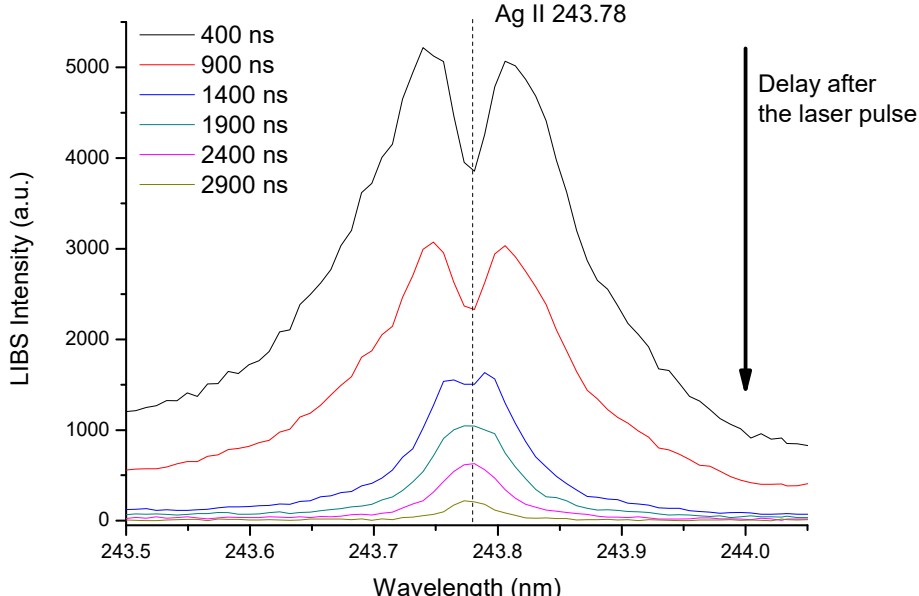

**Figure 2.** Self-reversal in a silver plasma. The different spectra were acquired at different delays after the onset of the plasma, with a gate of 500 ns. At later times, the plasma homogenizes, and the self-reversal disappears.

When the plasma is not homogeneous, moreover, even Equation (6) should be reconsidered, since we may have plasma conditions satisfying the McWhirter criterion, and at the same time not being in LTE conditions. The same may occur if the plasma is expanding too fast for the equilibrium to be established [39].

From the above discussion, it is clear the 'optimum' conditions for measuring spectral fundamental parameters by LIBS are the ones where Equations (1) and (3) hold. One should thus prefer experimental conditions leading to the creation of a homogeneous, stationary, optically thin plasma, in LTE conditions. In addition, the spectral signal/background ratio and its reproducibility should be high enough to guarantee a good quality of the results.

However, as we will see, sometimes it is not possible to fulfil all the above requirements. For example, the extent of the self-absorption effect in LIBS plasma is critical because, on one hand, Equation (1) holds only when the self-absorption of the plasma is negligible. On the other hand, self-absorption contributes to the establishment of the LTE conditions, therefore a balance between these two opposite requirements has to be found. Moreover, the expansion of the LIBS plasma depends on the type and density of the surrounding atmosphere, as well as on the laser pulse length, energy and focusing conditions on the target. The hypothesis of stationary and homogeneous plasma could be fulfilled only in a limited range of the LIBS plasma lifetime, or not being fulfilled at all. Similarly,

the strategy of using samples at low concentration of the element under study, to reduce the effect of self-absorption, may result in weak emission signals.

In the following, we discuss the experimental strategies used by LIBS researchers for determining fundamental spectral parameters (namely, line broadening parameters and transition probabilities) from the analysis of LIBS spectra.

## 3. Spectral Broadening and Shift Parameters

The main broadening effect in a LIBS plasma is the Stark effect [37]. It is due to the perturbation induced by the electric field of the electrons (and, at lesser extent, of the ions) on the energy levels of a given transition. The Stark broadening depends on the number density of the charged species (mainly the electron number density) and, at the same electron number density, it is in general larger for ions than for the neutral atoms.

This general model of Stark effect in a plasma can be summarized in the equation [43]:

$$\Delta\lambda_S = 2\, w(T, n_e)\left(\frac{n_e}{10^{16}}\right) + 3.5\, A \left(\frac{n_e}{10^{16}}\right)^{1/4}\left[1 - BN_D^{-1/3}\right]w(T, n_e)\left(\frac{n_e}{10^{16}}\right)$$
$$N_D = 1.72{\cdot}10^9\, T^{\frac{3}{2}}/n_e^{1/2}$$

(11)

where

$$B = \left\{\begin{array}{l} 0.75 \text{ for neutral lines} \\ 1.2 \text{ for ionic lines} \end{array}\right.$$

(12)

In Equation (11) $w(T, n_e)$ is the Stark broadening coefficient, depending weakly on the electron temperature and number density, A is the ion broadening parameter and $N_D$ represents the number of particles in the Debye sphere. The Stark effect is the main broadening phenomenon in LIBS plasmas (Doppler effect is typically negligible with respect to the Stark broadening). Moreover, In LIBS plasmas the ion contribution to Stark broadening can also be considered as negligible. Therefore, the FWHM of the Lorentzian line shape $L(\lambda)$ can be linked to the Stark broadening as:

$$\Delta\lambda_0 = 2\, w(T, n_e)\left(\frac{n_e}{10^{16}}\right)$$

(13)

For the ions, a shift in the position of the peak of the line can also be observed, depending on the plasma temperature and electron number density.

The simple relation between line width and electron number density is valid for all the elements, except hydrogen and hydrogenoid atoms. For this class of atoms, the Stark broadening is much more pronounced than in other elements; however, the dependence on the electron number density is also more complex [37,44]. An empirical formula linking the FWHM of the hydrogen Balmer alpha line (656.28 nm) to the electron number density has been derived in [45]:

$$\Delta\lambda_S = \gamma_\alpha(T, n_e)\, n_e^{2/3}$$

(14)

Similarly, Gigosos [44] proposed a formula for the FWHM of the hydrogen Balmer beta line (486.135 nm), as:

$$\Delta\lambda_S = \gamma_\beta(T, n_e)\, n_e^{0.68116}$$

(15)

$\gamma_\alpha(T, n_e)$ and $\gamma_\beta(T, n_e)$ depend weakly on the plasma temperature and electron number density. The description of the Stark broadening effect does not rely on the existence of LTE conditions in the plasma.

The measurement of the Stark broadening coefficient of a spectral line is, apparently, simple. Once the electron number density of the plasma is determined (from another line with known Stark

broadening coefficient—or from one of the hydrogen lines just mentioned [46], acquired at the same time and from the same spatial region) the Stark broadening coefficient could be determined as:

$$w(T, n_e) = \frac{\Delta\lambda_0}{2\left(\frac{n_e}{10^{16}}\right)} \tag{16}$$

The electron temperature can be determined, assuming LTE conditions in the plasma, using the usual Boltzmann [47] or Saha–Boltzmann [48] plot methods.

If the electron number density is determined from the Stark broadening $\Delta\lambda_{0ref}$ of another reference line whose Stark broadening coefficient $w_{ref}$ is known(not necessarily of the same species, since the electron number density is a global property of the plasma), the unknown Stark broadening coefficient can be simply calculated as:

$$w = w_{ref}\frac{\Delta\lambda_0}{\Delta\lambda_{0ref}} \tag{17}$$

without even measuring the electron number density. In this case, however, the dependence of the coefficient on the plasma temperature and electron number density cannot be made explicit.

There are, however, several obstacles to the direct application of the simple procedure envisaged in Equation (15) or Equation (16).

For the calculation of the Stark coefficient from the broadening the emission lines, one has to assume that:

(a) The plasma is homogeneous;
(b) The acquisition of the spectral signal is done in an interval short enough to maintain a reasonably constant electron number density;
(c) The Stark coefficient of the reference line is known with reasonable accuracy;
(d) The measured line widths (of the reference and unknown lines) are given only by the Stark effect.

While points (a), (b) and (c) are important, point (d) is evidently the key issue for the determination of the Stark coefficients of emission lines by LIBS. When factors other than Stark effect contribute to the measured line broadening, the Stark coefficient evaluated from Equation (15) would not be correct.

One obvious effect on the measured line broadening would be given by the instrumental broadening. When the spectral resolution of the spectrometer is comparable to the measured line broadening, its contribution could not be neglected. Moreover, the observed line shape would also change, from a pure Lorentzian to a Voigt profile, given by the convolution of the Lorentzian line with FWHM = $\Delta\lambda_0$ and a Gaussian with $\sigma$ = spectral resolution of the spectrometer:

$$V(\lambda) = \frac{1}{2\pi\sigma}\int_{-\infty}^{+\infty}\exp\left(-\frac{(x-\lambda)^2}{2\sigma^2}\right)\frac{\Delta\lambda_0}{(\lambda-\lambda_0)^2 + \left(\frac{\Delta\lambda_0}{2}\right)^2}dx \tag{18}$$

The two line widths (Gaussian and Lorentzian) could be, in principle, deconvolved by fitting the experimental line profile with a Voigt line shape.

However, since the resolution of the spectrometer can be obtained with good precision from the information provided by the manufacturer or determined by measuring the broadening of a narrow emission line, a more effective strategy would be directly measuring the FWHM of the experimental line profile with a Voigt function. After that, the Lorentzian component can be obtained by subtracting the experimental broadening from this value.

The subtraction should be performed correctly considering how the Gaussian and Lorentzian line widths contribute to the FWHM of the Voigt profile. Olivero et al. [49] proposed a formula for the Voigt FWHM that is correct within ±0.01%, as:

$$FWHM_V = FWHM_L \left( 0.5346 + 0.4654 \sqrt{1 + \left( \frac{FWHM_G}{0.4654\, FWHM_L} \right)^2} \right) \tag{19}$$

This relation can be inverted numerically for deriving the Lorentzian FWHM (the one related to the Stark effect) once the Voigt and the Gaussian FWHM are known.

The second problem, much more difficult to solve, is related to the self-absorption of the line, which introduces an apparent broadening of the line (see Figure 1) which increases with the increase of self-absorption.

Dealing with the self-absorption is made complicated by the fact that the line shape of a self-absorbed emission line is not described by a Lorentzian or Voigt profile anymore. Moreover, the presence of self-absorption sometimes is not easy to spot. One should consider, however, that even when the effect of self-absorption is not evident, the apparent FWHM of the line can be substantially larger than the Stark broadening (for $\tau(\lambda_0) = 1$, the apparent FWHM is about 30% larger than the unperturbed Stark width) [50].

The effects of self-absorption should thus be checked carefully, for avoiding errors in the determination of the Stark coefficients.

A (conceptually) simple method for determining experimentally the degree of self-absorption in LIBS plasma is the duplicating mirror method (see Figure 3) [51,52].

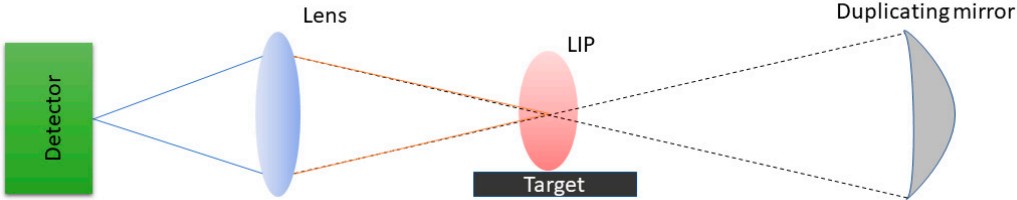

**Figure 3.** Duplicating mirror scheme.

The method assumes that the radiation emitted by the plasma in the direction opposite to the detector is sent back through the plasma, using a spherical mirror. By comparing the intensity of the line measured with and without the spherical mirror (ideally, one should be double of the other, in the absence of self-absorption effects) it is possible to evaluate the self-absorption of the plasma for that line. The measurement is not trivial, all the optical parts must be perfectly aligned and the reflectivity of the mirror, as well as all the other radiation losses, must be considered. However, this method allows a direct measurement of the plasma self-absorption and does not rely on any assumption on plasma homogeneity or close to LTE conditions, as the other methods discussed.

## 4. Transition Probabilities

Assuming a homogeneous and optically thin plasma, the spontaneous transition probability associated with the emission line could be, in principle, calculated using the branching ratio method, as:

$$A_{ji} = \frac{\beta_{ji}}{\tau_j} \tag{20}$$

where

$$\beta_{ji} = \frac{A_{ji}}{\sum_k A_{jk}} = \frac{I_{ji}}{\sum_k I_{jk}} \tag{21}$$

and $\tau_j$ is radiative lifetime of the upper level which can be measured or calculated theoretically.

When a reference emission line of the same species (with known transition probability) is available and the plasma is in LTE conditions, the unknown $A_{ki}$ coefficient can be determined using the relation:

$$A_{ki} = A_{kiref} \frac{g_{kref}}{g_k} \frac{I_0}{I_{0ref}} e^{-\frac{E_{kref} - E_k}{k_B T}} \tag{22}$$

Equation (21) is independent on the concentration of the species and, apparently, it provides a very simple and straightforward way for determining the unknown transition probabilities of the lines on interest.

The determination of the unknown transition probability can be made more precisely, exploiting the information that can be derived from the Boltzmann or Saha–Boltzmann plot for the corresponding species/element, as schematically represented in Figure 4.

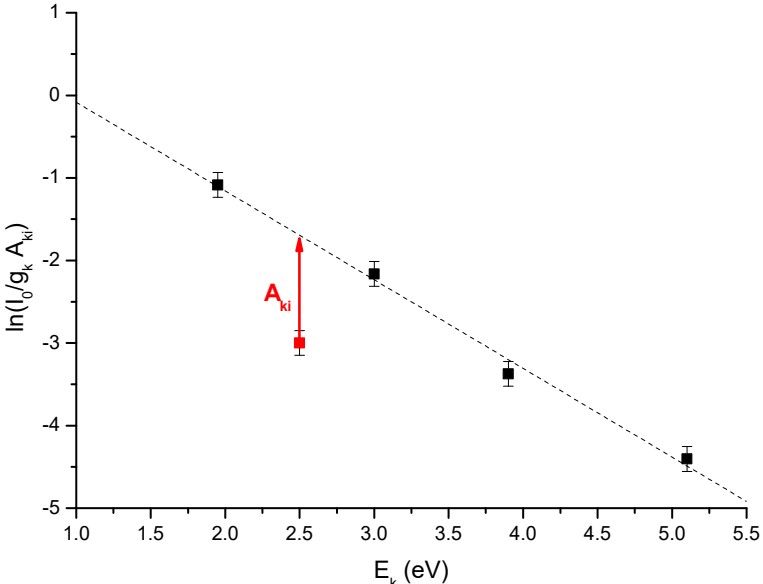

**Figure 4.** Determination of the unknown $A_{ki}$ of a spectral line from the Boltzmann plot of the same species.

Using a calibration-free LIBS approach [53], the unknown $A_{ki}$ can be determined even when another emission line of the same species is not available (or its $A_{ki}$ is not known). In this case, the plasma electron temperature is determined using the lines of another species (the temperature is a global parameter of the plasma and, in LTE conditions, is the same for all the species in the plasma). The knowledge of the composition of the sample is then used to determine the intercept of the curve, with slope $-1/k_B T$ for the species of interest (see Figure 5):

From the discussion on the determination of the Stark coefficients, it should be evident the limit of the above-discussed approaches.

For the application of these simple methods, the plasma must be, during the time of the measurements:

(a)    Homogeneous;
(b)    Stationary;
(c)    In LTE conditions; and, most of all
(d)    Optically thin (at least for the lines considered).

Equation (3), in fact, is valid in its form only when the self-absorption effects can be neglected. When this is not possible, the measured line intensity will be always lower than the corresponding $I_0$ in Equation (3), as described in Equation (8) for a line with Lorentzian line shape.

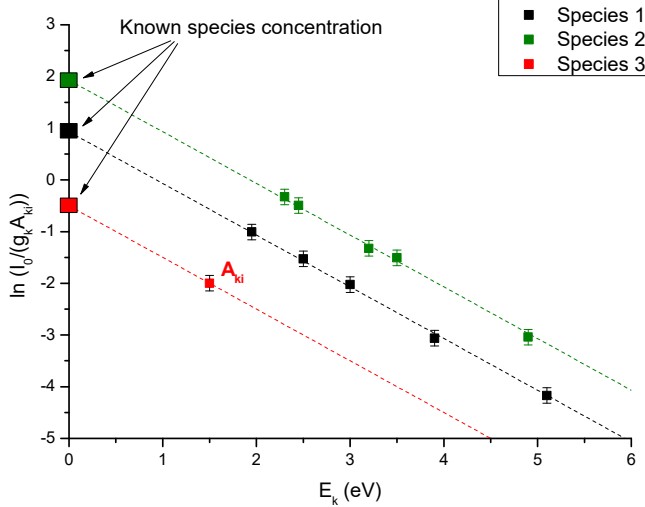

**Figure 5.** Determination of the unknown $A_{ki}$ of a spectral line from the knowledge of the composition of the sample.

The measured peak intensity of this line, after traveling a distance $l$ in the plasma, would be:

$$I(\lambda_0) = S(\lambda_o)\left(1 - e^{-k(\lambda_0)l}\right) \tag{23}$$

or equivalently:

$$I(\lambda_0) = SA \, I_0(\lambda_o) \tag{24}$$

where $I_0(\lambda_0)$ is the optically thin limit of the peak intensity, and the self-absorption parameter SA is defined as:

$$SA = \frac{1 - e^{-\tau(\lambda_0)}}{\tau(\lambda_0)} \tag{25}$$

*SA* is comprised between 1 ($\tau(\lambda_0) << 1$, negligible self-absorption) and 0 (infinite self-absorption). Note that, in the limit of infinite self-absorption, the peak line intensity is equal to the black-body intensity $S(\lambda_0)$ at the peak wavelength and temperature $T$ of the plasma, as shown in Figure 6.

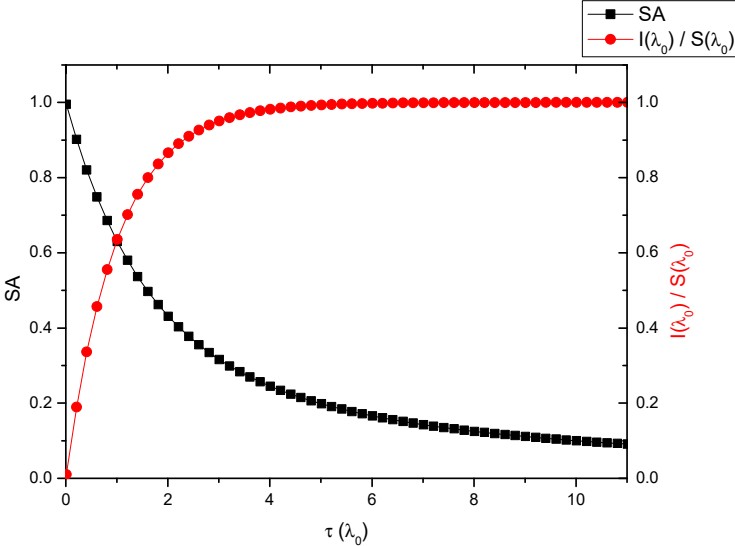

**Figure 6.** Dependence of *SA* (black) and normalized line peak intensity (red) on $\tau(\lambda_0)$.

The line peak intensity thus depends strongly on the plasma opacity. However, the Boltzmann equation (Equation (3)) is written in terms of the integral intensity of the emission line. The integral over the line profile described by Equation (8) in not analytical; however, it has been numerically solved by El Sherbini et al. [54], leading to a very simple relation with the *SA* parameter:

$$I = SA^{-0.54} I_0 \tag{26}$$

In the same work, the authors also determined the dependence of the line FWHM on *SA*, in the form:

$$\Delta\lambda = SA^{0.46}\Delta\lambda_0 \tag{27}$$

These two relations can be merged to obtain a link between measured intensity and FWHM of the emission line with the corresponding values in the optically thin limit [41]:

$$I = \left(\frac{\Delta\lambda}{\Delta\lambda_0}\right)^{-0.85} I_0 \tag{28}$$

Equation (26) is extremely useful for evaluating (and compensating) the effects of self-absorption. Using the above relation, Equation (3) can be rewritten in terms of the measured intensities as:

$$I\left(\frac{\Delta\lambda}{\Delta\lambda_0}\right)^{0.85} = Fn^a g_k A_{ki} \frac{e^{-\frac{E_k}{k_B T}}}{U^a(T)} \tag{29}$$

To link the measured intensities to the $A_{ki}$ spectral parameters, the optically thin FWHM of the lines ($\Delta\lambda_0$) must be known.

This link shows that in optically thick plasmas, the knowledge of the Stark coefficient of the emission lines of interest is essential for the determination of the $A_{ki}$, using either Equation (20) or the Boltzmann plot methods, unless the *SA* parameter can be evaluated with other independent methods. The application of the El Sherbini et al. relation requires the knowledge of the electron number density of the plasma, which can be obtained measuring the Stark broadening of a such as the hydrogen Balmer alpha line or another non-self-absorbed line whose Stark coefficient is known.

The incidence of self-absorption on the measured line intensity can be evaluated (and compensated) also using the dependence of the line intensity on the concentration of the corresponding emitting species. The *SA* parameter depends on the optical thickness of the plasma for the transition under study, and the optical thickness depends linearly on the concentration of the species $n^a$, according to Equations (9) and (10), as:

$$\tau(\lambda_0) = \frac{\lambda_0^4}{4\pi^2 c}\frac{1}{\Delta\lambda_0} A_{ki}\frac{g_k}{g_i}n^a \frac{e^{-\frac{E_i}{k_B T}}}{U^a(T)}l \tag{30}$$

If the electron temperature and electron number density of the plasma would not vary appreciably with the concentration of the element, the analysis of the LIBS line intensity vs. the element's concentration would provide information about the $A_{ki}$ or the $\Delta\lambda_0$ associated with the transition.

This idea is at the basis of the curve-of-growth method [55–57] and its variations, as the C-sigma method, introduced by Aragon and Aguilera in 2014 [58,59] and recently improved by Safi et al. [60].

According to the C-sigma method, the LIBS intensity of several lines and different species can be plotted on a universal curve, as a function of $\tau(\lambda_0)$. In the Safi et al. treatment (extended C-sigma), this universal curve is linear. Poggialini et al. [61] further improved the method, by introducing a Time-independent extended C-Sigma curve (TIECS) which does not depend on the time at which the LIBS spectrum is acquired.

The equation expressing the TIECS curve is [61]:

$$I\frac{1}{\Delta\lambda_0}\left(\frac{\Delta\lambda}{\Delta\lambda_0}\right)^{0.85} = \phi\, n^a g_k \frac{A_{ki}}{\Delta\lambda_0} \frac{e^{-\frac{E_k}{k_B T}}}{U^a(T)} \tag{31}$$

where *F* is a constant, which will be determined by the best fit of the experimental data. The RHS of Equation (30) is called C-sigma.

In all the version of the C-sigma curve method, once the universal curve is drawn using the data from non-self-absorbed lines (or lines with known Stark coefficient, from which the effects of self-absorption can be evaluated) the unknown parameters of another emission line (either $A_{ki}$ or $\Delta\lambda_0$) can be determined by adjusting the corresponding coordinates in expression (30) to lay on top of the curve, as shown in Figure 7.

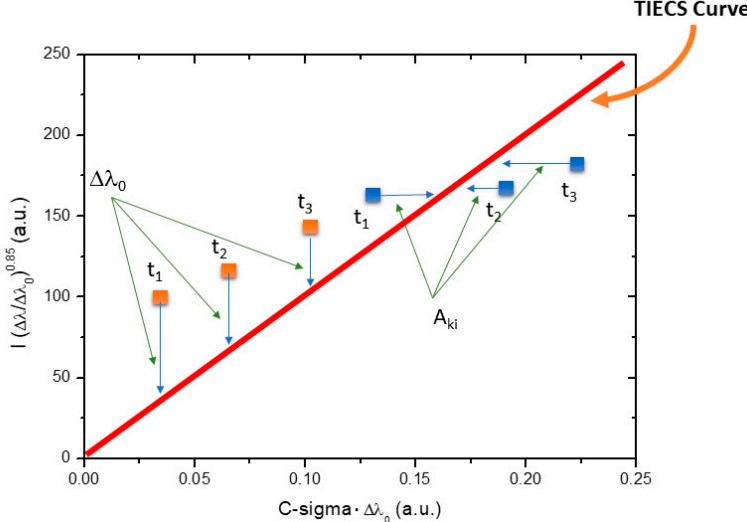

**Figure 7.** Determination of $\Delta\lambda_0$ or $A_{ki}$ from the TIECS curve.

Furthermore, in this case, for determining $n^a$ from the knowledge of the element's concentration in the sample and calculating the Stark coefficient from $\Delta\lambda_0$, the electron number density must be measured independently.

## 5. Experimental Determination of Spectral Parameters by LIBS

The main applications of LIBS to the determination of the fundamental spectral parameters of many emission lines of several elements, are summarized in two Tables (Table 1 for the Stark broadening coefficients and Table 2 for the transition probabilities), and briefly discussed in the following.

### 5.1. Determination of Stark Broadening Coefficients

Colón et al. [62] estimated, in 1993, the Stark broadening of 6 Al II lines during laser ablation of pure aluminum target in helium, argon and nitrogen. The authors found that 1000-mbar nitrogen pressure is an appropriate measurements condition for the broadening parameters. Plasma electron number density was determined using the AI II line at 466.305 nm. For this line, the Stark shift parameter was been also measured.

Chan et al. [63] in 1996 estimated the line widths of two doubly ionized aluminum lines. The experimental conditions used in this work permit to reach an electron number density higher than $10^{18}$ cm$^{-3}$ and plasma temperatures in the range of 3 to 7 eV. In these conditions, plasmas are considered optically thin. The authors found that their experimental results agree with the semi-empirical theoretical line width within the experimental errors of 20%.

**Table 1.** Applications of LIBS for determining the Stark broadening coefficients of atomic lines.

| Year | Authors | Ref. | Lines | Notes |
|------|---------|------|-------|-------|
| 1993 | Colón et al. | [62] | 6 Al II lines | |
| 1996 | Chan et al. | [63] | 2 Al III lines | |
| 1999 | Martinez and Blanco | [64] | 24 Sn II lines 3 Sn I lines | Self-absorption < 1% |
| 2004 | Cadwell et al. | [65] | 12 Ar I lines | Ar I line at 703.025 nm as a reference |
| 2005 | Ortiz and Mayo | [66] | 26 Au II lines | Abel inversion performed Self-absorption < 3% |
| 2006 | Bengoechea et al. | [67] | Fe I line at 381.58 nm | Self-absorption < 2% |
| 2006 | C. Colón and A. Alonso-Medina | [68] | 31 Pb II lines | Delay of 2.5 μs to reduce SA and calculation of $k(\lambda_0)l$ |
| 2007 | A. Alonso-Medina and C. Colón | [69] | 10 Pb III lines | Calculation of $k(\lambda_0)l$ |
| 2007 | Bredice et al. | [70] | 5 Mn I and 9 Mn II lines | Line width method |
| 2008 | R. Mayo et al. | [71] | 17 Ni II lines | Self-absorption < 2.5% |
| 2008 | R. Mayo and M. Ortiz | [72] | 6 Zn II lines | Calculation of $k(\lambda_0)l$ for the line 206.2 nm of Zn II and No dependence of the branching fractions on concentration in the 1–5% range of zinc content |
| 2008 | A. Alonso-Medina | [73] | 34 Pb I lines | Stark broadening parameters of the spectral lines were measured 2.5 μs after the laser pulse. |
| 2011 | A. Alonso-Medina | [74] | 25 Pb III lines | Calculation of $k(\lambda_0)l$ |
| 2011 | Aragon et al. | [75] | 21 Fe II lines | Curve-of-growth method |
| 2011 | Aguilera et al. | [76] | 26 Fe II lines | Curve-of-growth method |
| 2011 | Manrique et al. | [77] | | Curve-of-growth method |
| 2012 | Ferhat et al. | [78] | 6 Si II lines | Abel inversion |
| 2013 | Aguilera et al. | [79] | 53 Ni II lines | Curve-of-growth method |
| 2013 | El Sherbini et al. | [80] | 636.2 nm Zn I line | Line width method |
| 2013 | Cvejić et al. | [81] | 3 Mg I lines and 1 Mg II line | Duplicating mirror technique |
| 2014 | Aragon et al. | [82] | 36 Fe II and 27 Ni II lines | Curve-of-growth method |
| 2014 | Aragon et al. | [83] | 5 Fe II and 8 Ni II lines. | Curve-of-growth method |
| 2014 | Aguilera et al. | [84] | 11 Ca II lines | Curve-of-growth method |
| 2014 | Cirisan et al. | [85] | 4 Al II lines | Abel inversion for dealing with plasma inhomogeneity |
| 2015 | Nishijima et al. | [86] | 3 W I lines | |
| 2016 | Manrique et al. | [87] | 85 widths and 72 shifts of Ti II lines | Curve-of-growth method |
| 2016 | Popov et al. | [88] | 6 Mn I lines | Measurements on Fe–Mn alloys of different compositions |
| 2017 | Popov et al. | [89] | 3 Mn I lines | |

**Table 1.** *Cont.*

| Year | Authors | Ref. | Lines | Notes |
|------|---------|------|-------|-------|
| 2019 | A. Alonso-Medina | [90] | 2 Pb IV lines | Calculation of $k(\lambda_0)l$ |
| 2019 | A. Alonso-Medina | [91] | 2 Pb II lines | Calculation of $k(\lambda_0)l$ |
| 2019 | Dojic et al. | [92] | 18 Mo I and 18 Mo II lines | Duplicating mirror technique |
| 2019 | Manrique et al. | [93] | 41 Mn II lines widths and 30 Mn II lines shifts | Fused glass samples with varying Mn concentrations |
| 2020 | Dojić et al. | [94] | 5 Al II and 8 Al III lines. He I 388.86 nm line width and shift. | Duplicating mirror technique |
| 2020 | Poggialini et al. | [61] | 4 Ta I and 19 Ta II lines | TIECS method |

In 1999, B. Martinez and F. Blanco [64] determined the Stark broadening coefficients of 24 Sn II lines and three Sn I lines, in Ar atmosphere. The authors calculated the effect of self-absorption to be below 1%.

Cadwell et al. [65] in 2004 determined the Stark broadening coefficients of 12 Ar I lines, working in low-pressure conditions to avoid self-absorption of the radiation. They used the Ar I line at 703.025 nm as a reference line for the determination of the electron number density.

The next year, M. Ortiz and R. Mayo [66] calculated the Stark broadening coefficients of 26 lines of singly ionized gold. The measurements were done in Ar atmosphere, and the distribution of the emitters was checked using Abel inversion. The authors evaluated the effects of self-absorption and determined that it affected their calculation for less than 3%.

Bengoechea et al. [67] in 2006 studied on iron–nickel alloys the main experimental conditions that must be fulfilled by a LIBS plasma to determine the Stark broadening parameters. The authors found that the plasma inhomogeneity affects the Stark broadening determination with an error below 7%. The Stark broadening of the Fe I line at 381.58 nm was measured using the non-self-absorbed Fe I line at 538.34 nm as a reference for the determination of the electron number density. The effect of self-absorption for a sample with an iron concentration of 2%, was found to be negligible.

In the same year, C. Colón and A. Alonso-Medina [68] measured the Stark widths of 31 lines of Pb II. The LIBS experiments were performed on several lead targets, in vacuum and in argon atmosphere. Under 6 Torr of argon atmosphere, and at a delay of 2.5 μs after the laser, all the studied Pb II lines did not show self-absorption effects, except for the 220.3 nm line (this line was investigated separately by the same authors in 2019 [91]). After that (in 2007), using the same experimental setup A. Alonso-Medina and C. Colón [69] calculated the Stark widths of ten doubly ionized lead spectral lines (Pb III).

Experimental Stark broadenings of 5 Mn I and 9 Mn II lines were determined by Bredice et al. in 2007 [70]. Single and double-pulse LIBS configurations were used on a series of Fe–Mn alloy samples with Mn concentration ranging from 6% to 30%. The authors estimated the effect of self-absorption to be lower than 3%.

In 2008, R. Mayo et al. [71] determined the Stark widths of 17 Ni II lines. The measurements were done on binary Al–Ni alloy samples with low content of Ni (5% and 2%), in Ar atmosphere. For the most intense lines, the self-absorption factor calculations gave values lower than 2.5%; consequently, the plasma was considered optically thin.

In the same year, R. Mayo and M. Ortiz [72] studied a Cd–Zn alloy with a Zn content under 10%. The authors calculated the Stark widths of 6 lines of Zn II. To determine the SA effects, the authors found that there was no dependence of the branching ratios on concentration within the range of 5% to 1% of zinc in the sample.

A. Alonso-Medina measured by LIBS the Stark widths of 34 spectral lines of Pb I in 2008 [73] and 25 emission lines of Pb III in 2011 [74]. Plasmas were produced by laser ablation with a Nd:YAG laser on a Sn–Pb alloy (with a Pb content of about 0.5%) and lead target (99.99% purity) in an argon atmosphere. The author reported that self-absorption was lower than 3% [73] and 1% [74] for the most intense lines in the LIBS spectra, respectively.

In 2011, the experimental determination of the Stark width for 21 Fe II lines with wavelengths in the range 260–300 nm [75] and 26 Fe II lines in the range 230–260 nm [76] was done. The plasmas were generated from a set of Fe–Cu alloys (iron contents in the range 0.5 wt%–25 wt%) [75] and Fe–Cu/Fe–Ni samples [76], respectively. In these papers, the error due to self-absorption was estimated by authors to be lower than 10%.

Ferhat et al. [78] measured experimentally, in 2012, the Stark coefficients of 6 lines of Si II during pulsed laser ablation of a pure silicon target. A check of the temperature gradient along the radius of the plasma using an Abel inversion did not detect any inhomogeneity, and no self-absorption effects were detected.

In 2013, Aguilera et al. [79] estimated the Stark widths of 53 Ni II lines by laser ablation of Ni–Cu and Ni–Al alloys with a wide range of nickel concentrations. The authors used the curve-of-growth methodology on sample sets with different nickel concentrations to evaluate and reduce the effects of self-absorption. The error caused by self-absorption was estimated around 10%.

El Sherbini et al. [80] investigated by LIBS a ZnO target in air, to determine the Stark broadening of the atomic Zn I line at 636.2 nm. The self-absorption effects were evaluated using the method proposed by El Sherbini el al. [54].

The Stark widths of 3 Mg I and 1 Mg II lines were determined by Cvejić et al. [81] in a LIBS experiment on powder mixture of $Al_2O_3$, $Li_2CO_3$ and $MgCO_3$ sample. The authors aimed to reduce the Mg concentration in the sample to avoid strong self-absorption of the studied Mg I and Mg II lines.

In 2014, Aragon et al. [82,83] determined the Stark widths of 36 Fe II and 27 Ni II [82] and 5 Fe II and 8 Ni II spectral lines [83], respectively. In these two papers, several samples were investigated; fused glass samples with $Fe_2O_3$ and NiO oxides in powder form in the first [82] and Fe–Cu alloy and a $Fe_2O_3$ fused glass in the second study [83]. The effect of self-absorption on the Stark widths, gave an error of the Stark coefficients lower than 10%, according to the authors.

Stark widths and shifts of 11 Ca II lines were determined the same year by Aguilera et al. [84]. LIBS plasmas were generated by laser ablation of fused glass samples with different calcium concentrations, from 0.005 to 1 percent, prepared from pure calcium carbonate. The authors found that the self-absorption affected the Stark width determination with errors lower than 10%.

In another contribution Aguilera et al. [95] calculated 83 Stark widths and 49 shifts of Cr II lines. LIBS measurements were done on three fused glass samples prepared from pure chromium oxide with chromium atomic concentrations of 0.05, 0.1 and 0.2 percent. The error caused by neglecting self-absorption was estimated to be around 10%.

Cirisan et al. [85] in 2014 determined by LIBS the Stark broadening coefficient for 4 Al II lines. The electron number density was obtained through the measurement of the FWHM of the hydrogen Balmer alpha line. The author used the method of spatial Abel inversion to deal with the nonhomogeneity of the laser–plasma. The effect of self-absorption was calculated using the duplicating mirror method.

Nishijima and Doerner [86] reported in 2016 the first measurements of Stark broadening widths of W I lines (426.9 nm, 429.4 nm and 430.2 nm) as a function of the electron number density. The authors found a linear relation between W I Stark widths and electron number density. The authors calculated the electron number density values from the Stark broadening of a C II line at 426.7 nm in nanosecond laser-induced tungsten carbide plasmas.

The same year, Manrique et al. [87] measured 85 Stark widths and 72 shifts of Ti II lines. Plasmas were created during Nd:YAG laser ablation of fused glass discs with different titanium concentrations (0.1 and 0.5 percent). The self-absorption effects were estimated to be about 10%.

**Table 2.** Applications of LIBS for determining the transition probabilities of atomic lines.

| Year | Authors | Ref. | Lines | Notes |
|------|---------|------|-------|-------|
| 1993 | Gonzalez et al. | [96] | 38 Na II lines | |
| 1994 | Gonzalez et al. | [97] | 43 Cr II lines | |
| 1995 | Blanco et al. | [98] | 28 Si II lines | Self-absorption < 3% |
| 1995 | Ferrero et al. | [99] | 36 Ag II lines | Self-absorption < 5% |
| 1996 | Martinez et al. | [100] | 10 In II lines | Self-absorption < 1% |
| 1997 | Ferrero et al. | [101] | 59 Ni II lines | Self-absorption < 2.5% |
| 1997 | Alonso-Medina et al. | [102] | 30 Pb II lines | Calculation of $k(\lambda_0)l$ |
| 1997 | Gonzalez et al. | [103] | 32 Sb I lines | Calculation of $k(\lambda_0)l$ |
| 1999 | Colon et al. | [104] | 10 Pb III lines | Calculation of $k(\lambda_0)l$ |
| 1999 | Rojas et al. | [105] | 103 S II lines | Self-absorption < 1% |
| 2000 | Di Rocco et al. | [106] | 18 Xe II lines | The ratio of two lines of same upper level was used and confirmed negligible *SA* |
| 2000 | Alonso-Medina et al. | [107] | 36 Sn II lines. | Self-absorption < 3% |
| 2000 | Rojas et al. | [108] | 26 Sb III Lines | Self-absorption < 1% |
| 2001 | Matheron et al. | [109] | 24 Si II lines | Line width method for *SA* correction. |
| 2001 | Alonso-Medina et al. | [110] | 39 Pb I lines and Pb II at 220.35 nm | Self-absorption < 1% |
| 2004 | Ortiz et al. | [111] | 20 Ag II and 3 Cu II lines | Optimization of delay time at 200 ns. |
| 2005 | Mayo et al. | [112] | 120 Zr III lines | |
| 2005 | Biemont et al. | [113] | 40 Ag II and 40 Cu II lines. | |
| 2005 | Ortiz et al. | [114] | 2 Cd II and 2 Zr III lines | |
| 2005 | Mayo et al. | [115] | 38 Cd II lines | |
| 2007 | Mayo et al. | [116] | 16 Au II lines | No dependence of the branching ratios on concentration observed for gold content under 15% |
| 2010 | Alonso-Medina et al. | [117] | 30 Pb III lines | Calculation of $k(\lambda_0)l$ |
| 2010 | Alonso-Medina et al. | [118] | 97 Sn I lines | Calculation of $k(\lambda_0)l$ |
| 2011 | Manrique et al. | [77] | 19 Ni II lines | Curve-of-growth method |
| 2013 | Manrique et al. | [119] | 48 Ni II lines | Curve-of-growth method |
| 2013 | Ortiz et al. | [120] | 37 Re II lines | Calculation of $k(\lambda_0)l$ for 221.43 nm line of Re II |
| 2013 | Asghar et al. | [121] | 30 Ne I lines | |
| 2015 | Aguilera et al. | [59] | 9 Ca II lines | |
| 2015 | Mayo-García et al. | [122] | 112 oscillator strengths from Mo II lines | |
| 2016 | Aragon et al. | [123] | 161 oscillator strengths from Mo II lines | |
| 2018 | Manrique et al. | [124] | 46 Mn II lines | |
| 2019 | Urbina et al. | [125] | 6 W I lines | Time-resolved Boltzmann plot |
| 2020 | Alhijry et al. | [126] | 2 Ag I lines | Line width method. |

Furthermore, in 2016 Popov et al. [88] estimated the Stark widths of 6 Mn I lines. The LIBS experiments were conducted on an aluminum alloy target containing 80 ppm of manganese, to avoid strong self-absorption of Mn I lines. The same group in 2017 [89], determined the Stark widths and shifts of 3 Mn I lines in the range of temperatures $T = 7000$–9500 K and electron number densities $n_e$ between 4.8 and $45 \times 10^{16}$ cm$^{-3}$. In both works, they used a convex glass cylindrical lens for producing a long spark.

In 2019, A. Alonso-Medina acquired LIBS spectra of a pure lead target to determine the Stark widths of two Pb IV lines [90] and the Stark widths and shifts of the two ionic lines of Pb II at 220.35 nm and 438.65 nm [91]. In those papers, the authors confirmed the absence of important self-absorption effects after calculation of the absorption coefficient $k(\lambda_0)$.

The same year, Stark widths of 18 Mo I and 18 Mo II spectral lines were measured by Dojic et al. [92] by laser ablation of a molybdenum sample of high purity (99.9%). The authors used the duplicating mirror technique to evaluate the self-absorption effects.

Manrique et al. [93] in 2019 calculated the Stark widths of 41 Mn II lines and the Stark shifts of 30 Mn II lines. Fused glass samples with varying Mn concentrations (between 0.01 at% and 0.5 at%) were employed in this experiment. The effects of self-absorption were taken into account using the C-sigma graph method.

In 2020, Dojić et al. [94] investigated the Stark widths of Al II, Al III and He I 388.86-nm spectral lines. The measurement was done on a flat aluminum sample in a helium–hydrogen gas mixture at reduced pressure. The self-absorption was evaluated using the duplicating mirror technique.

Furthermore, in 2020, Poggialini et al. [61] used the time-independent extended C-sigma approach (TIECS) for the determination of the Stark broadening of 23 tantalum lines (4 neutral and 19 ionic). The LIBS plasma was generated by laser ablation of a pure tantalum foil of 250 µm thickness. The TIECS method intrinsically includes, in the calculation, the effects of self-absorption.

## 5.2. Determination of Transition Probabilities

In some papers, the oscillator strength $f$ of the transition is obtained, instead of the transition probability. The two parameters are equivalent, since:

$$f = 1.499 \times 10^{20} \, \lambda_0^2 \, \frac{g_k}{g_i} A_{ki} \tag{32}$$

where $\lambda_0$ is the peak wavelength of the transition (nm), $A_{ki}$ is the transition probability ($10^8$ s$^{-1}$) and $g_i$, $g_k$ are the degeneracies of the lower and the upper level of the considered transition, respectively.

Gonzalez et al. in 1993 [96], calculated transition probabilities of 38 Na II lines from measurements of emission-line intensities in a laser-produced plasma. The authors evaluated the effects of self-absorption by calculating the optical depth for the most intense Na II line (309.273 nm), showing that the plasma could be considered optically thin in their experiment.

The same group in 1994 [97] used the optical emission from a laser-produced plasma to measure the relative transition probabilities of 43 lines arising from the $3d^44p$ configuration of Cr II. Measurements were carried out with Al–Cr alloys with a Cr content of about 0.1%, to have an optically thin plasma. Transition probabilities were placed on an absolute scale by using, where possible, accurate experimental lifetimes from the literature and line-strength sum rules. The authors estimated the effects of self-absorption by calculating the absorption coefficient of the intense Cr II 206.542 nm line, demonstrating that for this line, self-absorption effects were negligible.

Blanco et al. [98] in 1995 determined the absolute transition probabilities of 28 Si II lines measuring the intensities of spectral lines from LIPs generated on a Silicon sample in Ar and Kr atmospheres. In this way, branching ratios of spectral lines arising from the $3s^24p$ levels were utilized to determine the transition probability values. Authors also calculated absolute transition probabilities of 23 Si II spectral lines using the Boltzmann plot method, under the assumption of local thermodynamic

equilibrium (LTE). A direct calculation indicates that self-absorption effects were below 3% for the strongest lines.

In the same year, Ferrero et al. [99] calculated the transition probabilities of 36 lines of Ag II using spectral lines emitted by a laser-induced plasma of pure silver sample. An estimation of the absorption coefficient was carried to verify that self-absorption was negligible. The authors found the absorption effects to be lower than 5% for the most intense lines.

Martinez et al. [100] in 1996 calculated the for the first time the absolute transition probabilities for 10 In II lines from branching-ratio measurements. In their experiments, the Indium sample was located inside a vacuum chamber, which was filled with argon at a pressure of 6 Torr. They found that the self-absorption corrections were below 1% for the strongest 468.11 nm line.

In 1997, Ferrero et al. [101] calculated the experimental transition probability values for 59 spectral lines of Ni II by measuring the emission line intensity of laser-produced plasmas of alloys of nickel and aluminum with low nickel content. To determine the transition probability values, branching ratios for the $3p^63d^84p$ to $3p^63d^84s$ lines of Ni II were measured. Some other transition probabilities were calculated by comparing their intensities with those of neighboring lines whose transition probabilities were previously determined. Relative values of transition probabilities were put on an absolute scale by using the lifetimes values. It was calculated that in their measurements, self-absorption effects were lower than 2.5% for the most intense lines.

In the same year, Alonso-Medina [102] used the laser-produced plasma of a lead target with 99.9% purity, to measure transition probabilities for 30 lines of Pb II. For nine of them, the results reported were the first experimental data obtained. The authors calculated the transition probabilities by plotting the intensities of the Pb II emission spectrum lines on a Boltzmann plot, assuming local thermodynamic equilibrium (LTE). To assess the effect of self-absorption, they calculated values of the optical depth not over 2%, thus confirming the optical thin hypothesis in their measurements.

González et al. [103] also in 1997 determined the experimental transition probabilities for 32 lines from the $5p^26s$ configuration of neutral Sb. All measurements were carried out with several Sn–Sb alloys were the Sb content was lower than 2% to have an optically thin plasma. The authors measured the emission intensities of lines arising from the same upper level to determine the relative transition probabilities and corresponding branching ratios. The results were placed on an absolute scale by using experimental lifetime values from the literature. To test the laser-produced plasma results, branching ratios have been obtained, where possible, by using a hollow-cathode lamp as a light source. It was found that the effect of self-absorption in the measurements for all spectral lines was lower than 4%.

In 1999, Colón et al. [104] determined the first experimental values of the atomic transition probabilities for ten spectral lines of Pb III from the measurement of emission-line intensities in an optically thin laser-produced plasma of Pb. Authors calculated relative transition probability values for lines of a given atomic species by measuring relative intensities, while absolute transition probabilities were obtained from relative line strength measurements by comparison with the relative line strength of the 560.89 nm Pb II transition in the Saha equation. They also considered the self-absorption effects in all measurements and determined that it affected their calculation for less than 3%.

Rojas et al. [105], in the same year, used a laser-produced plasma to measure the experimental transition probabilities for 103 lines from the $3s^23p^24p$ configuration of S II. The experiment was carried out with samples of zinc sulfide (ZnS) and antimony sulfide (Sb2S3) in a controlled atmosphere of pure argon. The authors obtained relative values from the measurement of emission-line intensities and the results were placed on an absolute scale by using line-strength sum rules and lifetime values. Moreover, they reported that the effects of self-absorption in their measurements were lower than 1.2%.

By using temporal spectra from a laser-induced plasma, Di Rocco et al. [106] in 2000 made an experimental and theoretical investigation on the transition probabilities of Xe II. They measured transition probabilities for eighteen lines by means of the Boltzmann plot method. The authors evaluated the effects of self-absorption by calculating the intensity ratio between the Xe II lines at

424.5 and 458.5 nm as a function of time. They found that the ratio remains almost constant about the mean value and thus, in that range, the plasma can be considered optically thin.

Furthermore, in 2000, Alonso-Medina and Colón [107] determined the transition probabilities values for 36 lines of Sn II by measuring the emission line intensity of a laser-produced plasma of lead and tin alloys. The transition probabilities were determined through the Boltzmann plot method and branching ratios measurements. The authors evaluated the effects of self-absorption on the measured lines by calculating the absorption coefficient and they demonstrated that for the most intense line, the self-absorption effect is less than 3%. The authors used the Boltzmann plot method to obtain absolute values of the transition probabilities.

In the same year, transition probabilities for 26 lines with origin from the $5s^2$(6p, 6f, 6d, 7s and 5g) configurations of Sb III were determined by Rojas et al. [108]. They used the branching ratios method and put the data on an absolute scale using calculated lifetimes and lines strengths. Measurements were carried out at different delays after the laser shot. The authors demonstrated that the effect of self-absorption is below 1%.

Matheron et al. [109] also in 2001 calculated the transition probabilities of 24 Si II lines. Measurements were done in air at atmospheric pressure by focusing a 10 ns laser beam ($\lambda$ = 1064 nm, $E$ = 140 mJ) on a silicon target, in a xenon and hydrogen atmosphere. In that work, the transition probabilities were deduced either from relative intensity ratios or by the branching ratio method. For correcting self-absorption effects, the authors used adapted numeric methods.

Alonso-Medina et al. [110] calculated the transition probability for the line 220.353 nm of Pb II and the transition probabilities for 39 lines of Pb I. In this case, they used Boltzmann plot method and branching-ratio measurements. To avoid self-absorption effects, the authors used Sn–Pb alloys with a lead content lower than 0.5%, in Argon atmosphere. They found that self-absorption effects were lower than 1% for the most intense lines; thus, the plasma could be considered optically thin.

In 2004, Ortiz et al. [111] used LIBS for determining the transition probabilities of 20 Ag II and 3 Cu II lines from the branching-ratio measurements obtained in the experiment. The measurements were carried out in a controlled Argon atmosphere (~8 Torr). The authors found a good agreement between the measured and the calculated transition probabilities values.

In another work, the same year Mayo et al. [112] determined by LIBS the transition probabilities of 120 emission lines of Zr III with 4d5p and 4d5d electron configurations. Their analysis was performed experimentally and theoretically on Cu–Zr alloy with a Zr concentration lower than 13% for maintaining an optically thin condition of plasma. Transition probabilities were estimated from branching ratio measurements and calculated lifetime magnitudes.

Biémont et al. [113] reported the experimental transition probabilities for Ag II spectral lines with electron configurations $4d^96s$ and $4d^95d$, and as well for Cu II lines from $3d^94d$ electron configurations. The authors used a Nd:YAG laser at 1064 nm wavelength to irradiate a silver or cooper target with 99.9999% purity in a controlled argon atmosphere (~8 Torr). The theoretical values of the transition probabilities were calculated by the Relativistic Hartree–Fock method considering configuration interaction and core-polarization influences. The absolute transition probabilities were measured taking into account the branching ratio and theoretical data related to radiative lifetimes.

Ortiz et al. [114] determined the transition probabilities and different plasma parameters (including temperature, composition and spectral self-absorption) of Cd II and Zr III, using a temporal and spatial resolved LIBS analysis.

Relative transition probabilities for 38 transitions arising from the $4d^{10}$(6p, 7p, 8p, 4f, 5f) and $4d^95s5p$ electron configurations of high lying states of Cd II were calculated by Mayo et al. [115] exploiting a LIBS plasma in LTE conditions. The experiment was performed on pure cadmium sample as well as on some Zn–Cd alloys, at Cd concentrations lower than 10%. The experimental transition probabilities were measured from branching ratios, which were converted to absolute values using theoretical data on the radiative lifetimes of corresponding states.

LIBS was used by Mayo et al. [116] in 2007 for the experimental determination of the transition probabilities of 16 Au II (all of them measured for the first time) with 6p configuration. The experiment was done on Cu–Au alloy with 10% Au content for achieving to the optically thin condition of plasma. Transition probabilities were evaluated experimentally by utilizing the theoretical lifetimes calculated in the same study.

Transition probabilities of 30 transitions belonging to $5d^{10}6s(8s, 7p, 5f, 5g)$ electron configurations of Pb III (20 measured for the first time) were experimentally obtained by Alonso-Medina [117] in 2010. A lead sample with a purity of 99.99% was irradiated with Nd:YAG laser at 1064 nm in a 6 Torr argon atmosphere so that no self-absorption effects were observed.

The same author [118] obtained experimentally the transition probabilities of 97 Sn I lines using a Pb–Sn sample. The relative values were put on an absolute scale using the available lifetimes of the upper states of the transition.

In 2011, the experimental determination of the Stark width for 21 Fe II lines with wavelengths in the range 260–300 nm [75] and 26 Fe II lines in the range 230–260 nm [76] was done. The plasmas were generated from a set of Fe–Cu alloys (iron contents in the range 0.5 wt%–25 wt%) [75] and Fe–Cu/Fe–Ni samples [76], respectively. In these papers, the error due to self-absorption was estimated by authors to be lower than 10%.

In another similar work, the same year Manrique et al. [77] obtained transition probabilities of 19 Ni II lines using the curve-of-growth method for taking into account the self-absorption effects in nickel samples with concentrations in a range between 0.11 at% and 27 at%.

Manrique et al. [119] exploited the optical emission spectra of the LIP generated by a Nd:YAG laser at 1064 nm wavelength to determine the transition probabilities of 48 emission lines of the transition array $3d^84s$–$3d^84p$ of Ni II (17 of which were measured for the first time). The authors used 12 home-made nickel–copper alloys samples with concentrations in the range between 0.11 at% and 27 at%. The self-absorption effects were compensated using the curve-of-growth method.

In 2013, Ortiz et al. [120] determined experimentally the transition probabilities for 37 spectral lines of Re II (15 of them were obtained for the first time) from a LIBS plasma produced by Nd:YAG laser irradiation of a low rhenium content Zn–Re alloy. The electron number density and temperature of the plasma were evaluated. The relative intensities were put on an absolute scale combining the branching ratios with the evaluation of the line lifetimes and by comparison with known spectral lines.

The same year, Asghar et al. [121] studied the spectral emission of a LIBS neon plasma produced by irradiation of a Nd:YAG laser at 1064 nm. The transition probabilities of 30 spectral lines of neon for transitions arising from $2p^53p$ upper and $2p^53s$ electron configurations were measured with the branching ratios method. The absolute values of the transition probabilities were obtained from the relative line strengths considering the lifetimes of the excited states.

In 2015, Aguilera et al. [56] applied the C-Sigma method for the determination of 9 Ca II transition probabilities (5 of which measured for the first time) from the 4d, 5s, 5d and 6s configurations. The authors used a fused glass (lithium borate) target added with high purity $CaCO_3$ and $Fe_2O_3$ for reference. Ca in the samples ranged from 0.07 at% to 1 at%.

Mayo-García et al. [122] in the same year experimentally determined 112 Mo II oscillator strengths (79 of which measured for the first time). The plasma was generated in air at atmospheric pressure from a fused glass target added with molybdenum oxide. Mo concentration was in the range of 0.1%. The relative intensities were reported on an absolute scale by considering the branching fractions ant the lifetimes of the upper levels of the transitions.

In 2016, Aguilera et al. [123] measured the oscillator strengths of highly excited levels of 161 Mo II lines (148 of which measured for the first time). The authors used fused glass samples added with molybdenum oxide.

In 2018, Manrique et al. [124] obtained the transition probabilities of 46 spectral lines of Mn II (28 of which measured for the first time) using the C-sigma graph method. They used glass disks prepared from pure MnO at different concentrations.

Urbina et al. [125] obtained in 2019 the atomic transition probability of six W I spectral lines using the 3D Boltzmann plot technique, a variation of classical Boltzmann plot which exploits the information from LIBS spectra taken at different delays to predict with a better accuracy the transition probabilities. All the spectral intensities were corrected using the El Sherbini method [54].

In 2020, Alhijry et al. [126] measured the transition probability of the two spectral lines of Ag I at 827.35 and 768.77 nm on silver targets irradiated by a Nd:YAG laser at 532 nm, using the Boltzmann plot method. The authors corrected the line intensities for self-absorption, using the El Sherbini method [54]. They reported transition probabilities two orders of magnitude lower with respect to the values in the literature.

## 6. Conclusions

The development of a solid knowledge on the effect of self-absorption on the measured line intensities and FWHM of the emission lines in a LIBS spectrum has opened the way to the use of the technique for precise determination of the fundamental spectral parameters (transition probabilities and Stark broadening coefficients) of several emission lines. This knowledge is of paramount importance for further LIBS analysis, as well as for many other spectrochemical applications.

**Author Contributions:** Conceptualization, V.P.; methodology, V.P., A.S., S.M.A.; data curation, S.M.A., A.S., A.B., B.C., S.L., F.P., S.R., F.R., V.P.; writing—original draft preparation, V.P., A.S., S.M.A., F.R.; writing—review and editing, S.M.A., A.S., A.B., B.C., S.L., F.P., S.R., F.R., V.P.; supervision, V.P.; funding acquisition, S.L. All authors have read and agreed to the published version of the manuscript.

**Funding:** Part of the activity was funded by a grant of the Italian Ministry of Foreign Affairs and International Cooperation (MACH project—RS19GR04).

**Conflicts of Interest:** The authors declare no conflict of interest.

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
