# Peer review of "Laser-Induced Breakdown Spectroscopy for Determination of Spectral Fundamental Parameters"

_applsci, doi:10.3390/app10144973_

Round 1
Reviewer 1 Report
This manuscript reviews the use of laser-induced breakdown spectroscopy (LIBS) for the determination of plasma and atomic parameters. The first part of the MS is devoted to the description of the techniques for the determination of the various parameters, and the second part gives a list of references to such works in the last three decades, including a very brief description of each.
The first part of the manuscript includes basic concepts and tools that can be found in most text books on plasma physics (e.g., Saha or Boltzmann Eqs.), as well as semi-empirical relations more specific to the field of LIBS. I believe that this information can be useful for physicists who wish to enter this field. The thorough work on the references will be useful for a larger community.
I recommend publications after some minor modifications. Below are my suggestions:
- Lines 54-55: mention also the problem of spatial gradients.
- Line 77: I find it awkward to write that the Boltzmann Eq. links the number of photons... . Boltzmann gives the population distribution. This distribution then can be used to link ...
- Fig. 2: Define "Delay"
- Eq. 11 requires a reference.
- Line 212: The authors should emphasize that the line under investigation and the reference line must be recorded simultaneously, from the same region.
- Eq. 16: Define subscript ref
- Some language-text polishing is required, e.g., lines 209, 413
Author Response
This manuscript reviews the use of laser-induced breakdown spectroscopy (LIBS) for the determination of plasma and atomic parameters. The first part of the MS is devoted to the description of the techniques for the determination of the various parameters, and the second part gives a list of references to such works in the last three decades, including a very brief description of each.
The first part of the manuscript includes basic concepts and tools that can be found in most text books on plasma physics (e.g., Saha or Boltzmann Eqs.), as well as semi-empirical relations more specific to the field of LIBS. I believe that this information can be useful for physicists who wish to enter this field. The thorough work on the references will be useful for a larger community.
I recommend publications after some minor modifications. Below are my suggestions:
- Lines 54-55: mention also the problem of spatial gradients.
This is an important point, we added a short discussion about it.
- Line 77: I find it awkward to write that the Boltzmann Eq. links the number of photons... . Boltzmann gives the population distribution. This distribution then can be used to link ...
This is correct, the equation is conventionally called ‘Boltzmann equation’ but it represents, indeed, the population of the upper level of the transition, given by the Boltzmann equation, times the transition probability. We clarified this point in the revised version of the manuscript.
- 2: Define "Delay"
We have modified the figure making explicit the delays after the laser pulse.
- 11 requires a reference.
Following the reviewer’s suggestion, we added a reference to Eq. (11)
- Line 212: The authors should emphasize that the line under investigation and the reference line must be recorded simultaneously, from the same region.
That’s true, we added a sentence about that in the revised manuscript
- 16: Define subscript ref
Done, we thank the reviewer for pointing out the missing definition.
- Some language-text polishing is required, e.g., lines 209, 413
We have rephrased the sentences at the lines signaled by the reviewer.
Reviewer 2 Report
This manuscript reviews of the works on applications of LIBS for the determination of the spectral fundamental parameters (line broadening parameters and transition probabilities) of atomic lines. The article gives the basic relationships describing the radiative processes in a laser-induced plasma. The article is provided with 125 references and a large table that includes links to specific examples of using LIBS to determine line broadening parameters or transition probabilities for various spectral lines of various elements. In my opinion, the collected information seems to be useful for professionals using LIBS for various applications.
However, the text of the manuscript requires substantial refinement in the sense of editing. The article cannot be published in this form. The authors write the names of physical quantities in a formula editor, and sometimes in a text editor, which leads to confusion. Many of the sentences are difficult to read. Particular attention should be paid to the text with formulas at lines 70 to 140. For instance, how should one understand the text with the formula (2), located at lines 74-75? In many places, authors ignore commas and periods, especially after formulas. I don’t understand how formulas (7) and (8) are related to each other? What is the unit of wavelength measurement on the abscissa in Fig. 1? Where does formula (11) come from? In Fig. 11, the name of the physical quantity must be indicated on the y-axis. What does “I(l0)/Black body” mean (see Figure 6)? Determination of Dl0 and Aki from the TIECS curve shown in Fig. 7 requires a more detailed explanation. Physical notations that are not defined in the manuscript are present on the coordinate axes of Fig. 7 (for example, what do b and a shown at the y-axis denote, and also what does the “*” sign used at the x-axis mean?).
Lines 392-679 provide a simple listing of things someone has measured and the points in time when it happened. However, there is no generalization of the results obtained by other authors in the text of the manuscript.

Author Response
This manuscript reviews of the works on applications of LIBS for the determination of the spectral fundamental parameters (line broadening parameters and transition probabilities) of atomic lines. The article gives the basic relationships describing the radiative processes in a laser-induced plasma. The article is provided with 125 references and a large table that includes links to specific examples of using LIBS to determine line broadening parameters or transition probabilities for various spectral lines of various elements. In my opinion, the collected information seems to be useful for professionals using LIBS for various applications. However, the text of the manuscript requires substantial refinement in the sense of editing. The article cannot be published in this form. The authors write the names of physical quantities in a formula editor, and sometimes in a text editor, which leads to confusion.
Many of the sentences are difficult to read. Particular attention should be paid to the text with formulas at lines 70 to 140. For instance, how should one understand the text with the formula (2), located at lines 74-75? In many places, authors ignore commas and periods, especially after formulas.
We thank the reviewer for pointing out the problems in the formulas. They were written using Microsoft Equation Editor and, honestly, this is the first time that such problems occur. It seems that the template provided is not compatible with the Equation Editor, and all the formulas are converted to images. We agree with the reviewer that the result is awful. We did our best to improve the readability of the manuscript, in the revision.
I don’t understand how formulas (7) and (8) are related to each other? What is the unit of wavelength measurement on the abscissa in Fig. 1?
We have tried to explain better this point. Eq. (7) is the ‘optically thin’ line shape, eq. (8) represents the ‘optically thick’ expression, the Lorentzian line shape is distorted as shown in figure 1. The figure was just representative, but we have modified it to comply with the reviewer’s suggestion. We reorganized the text and the equation for better clarity.
Where does formula (11) come from?
Following the reviewer’s suggestion, we added a reference to Eq. (11)
In Fig. 11, the name of the physical quantity must be indicated on the y-axis.
In the manuscript, there are only 7 figures. Is the reviewer referring to figure 6? The y-axis represents the value of the ratios in the caption. It’s a numerical quantity. We have redrawn the figure with a double axis to make it clearer.
What does “I(l0)/Black body” mean (see Figure 6)?
We have rewritten this part of the caption as
Determination of Dl0 and Aki from the TIECS curve shown in Fig. 7 requires a more detailed explanation. Physical notations that are not defined in the manuscript are present on the coordinate axes of Fig. 7 (for example, what do b and a shown at the y-axis denote, and also what does the “*” sign used at the x-axis mean?).
We have modified figure 7 for making it clearer (we substituted the * with ∙ as symbol of multiplication), and we replaced in the figure the ratio between b and a with its numerical value (-0.85). We have also identified C-sigma with the RHS of eq. (30).
Lines 392-679 provide a simple listing of things someone has measured and the points in time when it happened. However, there is no generalization of the results obtained by other authors in the text of the manuscript.
We have added a brief discussion on the methods traditionally used for the determination of Aki and Stark coefficients and briefly discussed the benefits and drawbacks of LIBS as plasma source.
Reviewer 3 Report
This is a very solid contribution in the field of laser spectroscopy, and more precisely Laser-Induced Breakdown Spectroscopy (LIBS). The paper discusses the application of the LIBS technique to the determination of plasma emission fundamental parameters.
When reading the last part of the manuscript it really feels like an expert-written review paper - a lot of data and many lines of critical discussions on other (similar to some extent) contributions.
The only problem I have with this manuscript is that it's so highly specialized, so it is in some places hard to follow even for someone working in the laser spectroscopy field (like myself).
Author Response
This is a very solid contribution in the field of laser spectroscopy, and more precisely Laser-Induced Breakdown Spectroscopy (LIBS). The paper discusses the application of the LIBS technique to the determination of plasma emission fundamental parameters.
When reading the last part of the manuscript it really feels like an expert-written review paper - a lot of data and many lines of critical discussions on other (similar to some extent) contributions.
The only problem I have with this manuscript is that it's so highly specialized, so it is in some places hard to follow even for someone working in the laser spectroscopy field (like myself).
We warmly thank the reviewer for appreciating our work.
Reviewer 4 Report
The manuscript devoted to the special issue "Laser Induced Plasma/Breakdown Spectroscopy". This Special Issue aims to collect original research papers and REVIEWS ON RECENT METHODOLOGICAL AND TECHNOLOGICAL DEVELOPMENTS and/or novel uses of LIPS/LIBS in its broad applicative domains. The main keywords are quantitative analysis, chemometrics, elemental mapping, LIPS spectra processing, portable instrument development, trace analysis, in-situ measurements, environmental and geological analysis, industrial applications, biomedical and pharmaceutical applications, cultural heritage analysis. Connection between content of this manuscript and recent methodological and technological developments of LIBS is poor. The text contains a lot of basic well-known theory. In turn, analysis of «experimental strategies used by several authors for the determination of the spectral experimental parameters using LIBS» is poor. Unfortunately, I can’t recommend this manuscript to the publication in the present form.
Main comments.
- Authors should expand the abstract and give a part of the conclusions made. It is necessary to reflect the usefulness of this review.
- Chapter 5. lines 385-679. Most of the text presented can be obtained from the abstracts of these articles. I recommend rewrite this section. Table 1 is inconvenient. There are several options for better sorting its content instead of chronological order. For example, by types of atoms, by groups of researchers, or by types of determined spectral characteristics. In this case, the reader will be able to quickly find the necessary information.
Additional comments.
- Line 74. «(cm) is the thermal wavelength of the electron». Check and fix it, please.
- 131 line (figure 1). Dimension of «wavelength» is not clear.
Author Response
The manuscript devoted to the special issue "Laser Induced Plasma/Breakdown Spectroscopy". This Special Issue aims to collect original research papers and REVIEWS ON RECENT METHODOLOGICAL AND TECHNOLOGICAL DEVELOPMENTS and/or novel uses of LIPS/LIBS in its broad applicative domains. The main keywords are quantitative analysis, chemometrics, elemental mapping, LIPS spectra processing, portable instrument development, trace analysis, in-situ measurements, environmental and geological analysis, industrial applications, biomedical and pharmaceutical applications, cultural heritage analysis. Connection between content of this manuscript and recent methodological and technological developments of LIBS is poor.
We don’t agree with the reviewer on this point. The topic of this invited review was agreed with the Editor to offer to the readers something which is not usually found in thematic issues but could be very useful for LIBS researchers. In fact, even not considering the growing number of papers that have been published in the last few years on the determination of fundamental spectroscopic parameters by LIBS, it should not be necessary to stress that their knowledge is of paramount importance for many of the applications of LIBS that the reviewer considers more interesting, in particular for quantitative analysis by Calibration-Free LIBS and for plasma diagnostics, just to give two examples.
The text contains a lot of basic well-known theory.
The theoretical background that we have summarized in the first part of the manuscript is not complex, in fact. However, we’d rather not say that the theory is ‘well-known’, since the basic concepts that we have presented are often ignored, or misused, in practical applications.
In turn, analysis of «experimental strategies used by several authors for the determination of the spectral experimental parameters using LIBS» is poor.
We apologize, but we don’t understand the meaning of the last remark of the reviewer. In any case, we have modified the sentence, supposing that the remark was a suggestion for a better formulation of the phrase.
Main comments.
- Authors should expand the abstract and give a part of the conclusions made. It is necessary to reflect the usefulness of this review.
We have expanded the abstract, according to the reviewer’s suggestion.
- Chapter 5. lines 385-679. Most of the text presented can be obtained from the abstracts of these articles. I recommend rewrite this section.
This section is hard to further summarize, and could not be expanded, because of the large number of papers to be discussed. However, the purpose of a review is to give the most significant result of the relevant paper, and this is what we have tried to do.
- Table 1 is inconvenient. There are several options for better sorting its content instead of chronological order. For example, by types of atoms, by groups of researchers, or by types of determined spectral characteristics. In this case, the reader will be able to quickly find the necessary information.
Yes, indeed. The sorting method is mostly a matter of personal taste; we could have sorted the Table by the surnames of the first authors, or by research groups. But the first authors can change in the publications of the same group, or the composition of the group itself can change. Also sorting the table by element would have been difficult, since several papers calculated the parameters of more than a single element. Chronological order seemed to us the most effective for the reader to do a quick research of recent papers or referring to earlier works, if needed. At the end, the number of publications is large, but no so large to make difficult a cursory search on the relevant columns. In any case, to follow the reviewer’s suggestion, in the revised version of our manuscript we have divided the table and the corresponding discussion in two parts (Stark broadening coefficients and Aki). In this way the discussion would be also more punctual to the corresponding reference.
Additional comments.
- Line 74. «(cm) is the thermal wavelength of the electron». Check and fix it, please.
Thank you very much for pointing out this error. It seems that the template provided has problems with formulas and symbols. We have corrected the issue.
- 131 line (figure 1). Dimension of «wavelength» is not clear.
The figure was representing an ideal case. We have redrawn it, specifying that the units are arbitrary.
Round 2
Reviewer 4 Report
Authors gave answers to my comments. The manuscript can be accepted.